# The Impact of Blood Transfusion in Developing Postoperative Delirium in Patients with Hip Fracture Surgery

**DOI:** 10.3390/jcm12144696

**Published:** 2023-07-14

**Authors:** Sang-Soo Lee, Jong-Ho Kim, Jae-Jun Lee, Young-Suk Kwon, Eun-Min Seo

**Affiliations:** 1Department of Orthopedic Surgery, Chuncheon Sacred Heart Hospital, Hallym University College of Medicine, Chuncheon 24253, Republic of Korea; sslee123@gmail.com; 2Division of Big Data and Artificial Intelligence, Institute of New Frontier Research, Chuncheon Sacred Heart Hospital, Hallym University College of Medicine, Chuncheon 24253, Republic of Korea; poik99@hallym.or.kr (J.-H.K.); iloveu59@hallym.or.kr (J.-J.L.); 3Department of Anesthesiology and Pain Medicine, College of Medicine, Chuncheon Sacred Heart Hospital, Hallym University College of Medicine, Chuncheon 24253, Republic of Korea

**Keywords:** postoperative delirium, hip fracture surgery, clinical data warehouse, risk factor, blood transfusion

## Abstract

Background: Many studies have been conducted to explore the risk factors associated with postoperative delirium (POD) in order to understand its underlying causes and develop prevention strategies, especially for hip fracture surgery. However, the relationship between blood transfusion and POD has been heatedly debated. The purpose of this study was to evaluate the risk factors of POD and the relationship between blood transfusions and the occurrence of POD in hip fracture surgery through big data analysis. Methods: Medical data (including medication history, clinical and laboratory findings, and perioperative variables) were acquired from the clinical data warehouse (CDW) of the five hospitals of Hallym University Medical Center and were compared between patients without POD and with POD. Results: The occurrence of POD was 18.7% (228 of 2398 patients). The risk factors of POD included old age (OR 4.38, 95% CI 2.77–6.91; *p* < 0.001), American Society of Anesthesiology physical status > 2 (OR 1.84 95% CI 1.4–2.42; *p* < 0.001), dementia (OR 1.99, 95% CI 1.53–2.6; *p* < 0.001), steroid (OR 0.53 95% CI 0.34–0.82; *p* < 0.001), Antihistamine (OR 1.53 95% CI 1.19–1.96; *p* < 0.001), and postoperative erythrocyte sedimentation rate (mm/h) (OR 0.97 95% CI 0.97–0.98; *p* < 0.001) in multivariate logistic regression analysis. The postoperative transfusion (OR 2.53, 95% CI 1.88–3.41; *p* < 0.001) had a significant effect on the incidence of POD. Conclusions: big data analytics using a CDW was a good option to identify the risk factors of POD and to prevent POD in hip fracture surgery.

## 1. Introduction

Postoperative delirium (POD) is often a complication in patients with hip fracture surgery [1]. The occurrence of POD after hip fracture surgery varied from 9 to 42% [2]. POD is involved with a long hospital stay, higher financial costs, increased morbidity, and mortality [3]. In previous studies, several strategies (including the identification of risk factors for POD) have been studied to identify the pathogenesis of POD and prevent POD in hip fracture surgery [4,5,6]. However, the pathogenesis remains controversial. One of the mechanisms is a deficit in brain perfusion, which is caused by hypotensive events in relation to intraoperative blood loss [7,8]. Therefore, blood transfusion can be associated with the development of POD after hip fracture surgery [9,10]. Multiple studies have investigated the relationship between blood transfusion and POD in hip fracture surgery. Some studies have reported a positive relationship between blood transfusion and POD [3,11,12], while other studies have reported inconsistent findings (no relationship) [13,14].

Recently, big data analysis using a clinical data warehouse (CDW) has been used in medical research. This has made it possible to analyze structured and unstructured data quickly and precisely [15]. Hence, we applied big data analysis using a CDW to evaluate the incidence and risk factors of POD quickly and precisely.

Therefore, the purpose of this study was to identify the risk factors of POD and evaluate the relationship between blood transfusions and the occurrence of POD in hip fracture surgery through big data analysis.

## 2. Materials and Methods

### 2.1. Ethics and Data

This study is a retrospective study and was performed after achieving approval from the Clinical Research Ethics Committee of Hallym University Chuncheon Sacred Heart Hospital. Although this study included vulnerable participants such as the elderly, informed consent was waived for all subjects because patients had already completed their treatment, and this was a historical cohort analysis of the clinical data obtained from the completed course of treatment.

All data were obtained from the CDW of Hallym University Medical Center. CDW included clinical data from 5 hospitals of Hallym University Medical Center, and the clinical data included an integrated database of outpatient and inpatient medical records, prescriptions, and laboratory and imaging tests.

### 2.2. Participants

The participants were patients >18 years of age who underwent hip surgery due to the fracture of their hip joint from January 2011 to July 2022. Exclusion criteria were as follows: patients had delirium before surgery, patients underwent ventilator treatment after surgery, patients experienced a lack of consciousness after surgery, patients underwent other surgery simultaneously, patients underwent sedation therapy after surgery, and patients had missing data.

### 2.3. Exposure Variables and Primary Outcomes

The exposure variable included the intraoperative transfusion amount of red blood cells (RBC) and the transfusion of RBC during observation periods after surgery. Pack and Pack/day were used as the unit for the intraoperative transfusion and postoperative transfusion, respectively. The occurrence of POD was evaluated for 7 days after surgery. The short-form Korean Nursing Delirium Screening Scale was used for the diagnosis of POD. Once patients were suspected to be POD through the short-form Korean Nursing Delirium Screening Scale, the patients were referred for psychiatric counseling, and the diagnosis of POD was confirmed. The occurrence of POD was defined as patients with words indicating the specific symptoms and signs of delirium in postoperative psychiatric counseling notes. The list of words indicating specific symptoms and signs of delirium was predetermined in a previous study (Appendix A) [16]. Additionally, the odds ratios of perioperative risk factors, including intra- and postoperative red blood cell transfusion, were calculated.

### 2.4. Preoperative Factors

The preoperative risk factors were: age, male, obesity (body mass index >29.9), an American society of Anesthesiology physical status > 2, emergency surgery, preoperative morbidities(hypertension, diabetes, stroke, cancer, Parkinson disease, dementia, depression, kidney disease, liver disease, insomnia), medication (calcium channel blocker, diuretics, beta blocker, angiotensin converting enzyme inhibitors, angiotensin-receptor blockers, miscellaneous drug antidepressants, hypnotics, anti-psychotics, opioid, non-steroidal anti-inflammatory drug, analgesic except non-steroidal anti-inflammatory drug, muscle relaxants, steroid, anti-platelet, anti-coagulants, anti-hyperlipidemic, anti-Parkinson, antihistamine, genitourinary drug, h2-blocker), alcohol, smoking, and a laboratory test (hemoglobin [≥10, 9.9–8, 7.9–6, <6 g/dL], aspartate aminotransferase [difference from upper normal level], alanine aminotransferase [difference from upper normal level], sodium [hyponatremia, normal level, hypernatremia], potassium [hypokalemia, normal range, hyperkalemia], uric acid [difference from upper normal range], blood urea nitrogen [difference from upper normal range], creatinine [difference from upper normal range], albumin [hypoalbuminemia, normal level, hyperalbuminemia]). The list of drugs was created in a previous study (Appendix B) [16].

### 2.5. Intraoperative Factors

Intraoperative risk factors included the surgical type (open reduction and internal fixation, hip hemiarthroplasty, and total hip arthroplasty), operation time, the use of propofol as maintenance anesthetics, the use of opioids, amounts of midazolam (mg), oliguria (<0.5 mL/kg/h), administered fluid amounts (L), estimated blood loss (L), and the transfusion amount(pack).

### 2.6. Postoperative Factors

Postoperative risk factors included the transfusion amount, laboratory test (hemoglobin, sodium, potassium, uric acid, blood urea nitrogen, creatinine, albumin, aspartate aminotransferase, alanine aminotransferase, C-reactive protein, erythrocyte sedimentation rate), maximum body temperature, intensive care unit admission, and patient-controlled analgesia. Postoperative laboratory test results were expressed as the same method as preoperative laboratory tests.

### 2.7. Statistics

The statistical analysis was performed by IBM SPSS Statistics (version 26.0; IBM Corp., Armonk, NY, USA). Continuous data were described as median, interquartile ranges (IQRs), and maximum values, while categorical data were described as frequencies and percentages. Continuous data were analyzed by the Mann–Whitney test or independent *t*-test, and categorical data were analyzed by the chi-square test. The unadjusted and adjusted odds ratio and 95% confidence intervals were calculated to develop POD using Multivariate Coxregression. Statistical significance (*p*-value < 0.05) was considered.

## 3. Results

From January 2011 to July 2022, 6984 patients underwent hip surgery due to hip fractures at one of the five hospitals of Hallym University. In total, 4586 patients were excluded. Thus, 2398 patients were initially included in this study. The flow chart is summarized in Figure 1.

After surgery, 448 (18.7%) patients developed POD within 7 days. The preoperative, intraoperative, and postoperative risk factors of patients are summarized in Table 1.

Intraoperative red blood cell transfusions were performed for 226 (50.4%) of the patients who experienced POD and 810 (41.5%) of the patients without POD. Postoperative red blood cell transfusions were performed for 158 (35.3%) of the patients who experienced POD and 878 (45.0%) of the patients without POD. The red blood cell transfusion distributions are summarized in Table 2.

The hazard ratio of the intraoperative red blood cell transfusion and postoperative red blood cell transfusion volume are summarized in Table 3. The odds ratios of other factors for developing POD are summarized in Table 4.

## 4. Discussion

Anemia has been known to be a risk factor for POD after hip fracture surgery [11,13]. Preoperative anemia is common in patients with hip fractures due to fracture site bone bleeding. The occurrence of preoperative anemia was higher than 60% in patients with a hip fracture. Prolonged hypotension could lead to low cerebral perfusion, cerebral ischemia, hypoxia, and the impairment of brain function [17,18]. In recent years, a newmonitoring tool using artificial intelligence has been developed, which can predict episodes of intraoperative hypotension and prevent prolonged hypotension [19,20]. Perioperative blood loss can also lead to decreased cerebral blood flow (cerebral ischemia, hypoxia, impairment of brain function), impaired metabolism, and inflammatory reactions [17]. The impairment of oxygen delivery caused by low cerebral perfusion could cause POD [18]. If large blood losses lead to unstable vital signs in patients, a red blood cell transfusion is conducted. Therefore, red blood cell transfusion due to perioperative blood loss is highly associated with POD. Several investigators have explored the association between red blood cell transfusion and POD after hip fracture surgery; however, these results are controversial. Some studies have reported a positive association between red blood cell transfusion and POD [21,22,23], while others have found no association [14,24]. In this study, there was no relationship between the pre-, intra-, and postoperative hemoglobin level (anemia) with red blood cell transfusion and the incidence of POD after hip fracture surgery. However, postoperative red blood cell transfusion has also been associated with developing POD. These findings suggest that the perioperative hemoglobin level is not related to POD. Therefore, red blood cell transfusion-related complications (like increased inflammation and oxidative stress) might contribute to the occurrence of POD [25,26]. One unit of the red blood cell has a million donor leukocytes and proteins. Allogenic leukocytes or proteins could enhance inflammation and injure immunity. In a recent study, an older red blood cell transfusion was associated with increased neuro-inflammation and the increased duration of delirium [27,28]. The optimal transfusion trigger in orthopedic surgical patients recommends maintaining a hemoglobin level of >8.0 g/dL [14]. However, some studies compared red blood cell transfusion (to maintain a hemoglobin level of >8.0 g/dL) with restrictive red blood cell transfusion (to maintain a hemoglobin level of >7.0 g/dL). A restrictive red blood cell transfusion significantly decreased cardiac morbidity and readmission in hip fracture surgery [29]. Therefore, a restrictive red blood cell transfusion has been recommended to prevent POD in hip fracture surgery.

The occurrence of POD after hip fracture surgery varied between 12% and 51% [9,30]. In this study, the occurrence of POD was 18.7%. The differences between diagnostic methods, surgery procedures, and anesthesia could lead to a variety in the occurrence of POD [21,31,32].

POD is a serious complication after surgery in patients. POD has been shown to delay recovery and prolong hospital stays [33,34]. Therefore, the identification of the risk factors for POD is the first step to clarifying the pathogenesis of POD and preventing POD [30,35]. In this study, the risk factors of POD were identified in hip fracture surgery through a complete evaluation using big data. Old age, American Society of Anesthesiologists physical status >2, dementia, antihistamine, steroid, erythrocyte sedimentation rate(mm/h), and postoperative red blood cell transfusion were significantly associated with POD (Table 4).

Previous studies have shown that the higher occurrence of POD in old age could be related to increased comorbidities as well as impaired physiologic compensatory capability to adjust surgical stress [30]. In addition, the changes in the content of central neurotransmitters (such as acetylcholine, norepinephrine, epinephrine, etc.) were associated with POD in advanced age [22].

Some studies have reported that the type of anesthesia is a significant risk factor for POD [23]. Additionally, inconsistent findings have been reported [36,37]. In this study, there was no significant difference in the type of anesthesia between patients with POD and those without POD.

Several studies have shown that ASA status is linked to poor general conditions and multiple comorbidities [38]. Additionally, ASA classification (>2 levels) was associated with POD after hip fracture surgery in this study.

Generally, polypharmacy, medications with anti-psychotics, and anticholinergic activity have been associated with POD [24]. All drugs were investigated. In this study, the administration of antihistamines could be related to the higher incidence of POD. On the other hand, the administration of the steroid could be related to the lower incidence of POD.

Emergency surgery makes patients feel uncomfortable. Emergency surgery has been related to the occurrence of POD [39]. However, other studies reported a higher occurrence of POD in non-emergency surgeries [40]. In this study, there was no significant relationship between emergency surgery and POD.

This study has some limitations. The short-form Korean Nursing Delirium Screening Scale was used for the diagnosis of POD. Once patients were suspected to have POD through the short-form Korean Nursing Delirium Screening Scale, the patients were referred to psychiatric counseling, and the diagnosis of POD was confirmed. Therefore, patients who had mild symptoms of POD and did not receive treatment by psychiatrists could have been excluded. The exclusion of these patients might have affected the accuracy of the diagnosis of POD and the analysis of the association among risk factors.

Another limitation was that the occurrence of POD was evaluated 7 days after hip fracture surgery. Patients who developed POD after discharge were not included. The exclusion of these patients could introduce selection bias and limit the representativeness of the sample.

All confounding factors influencing the relationship between blood transfusion and POD could not be adjusted due to the observational nature of this study. This could have affected the accuracy of the causality evaluation. Finally, data were retrospectively analyzed. Therefore, the brain imaging study was not evaluated in all patients. The severity of POD and cognitive dysfunction could not be analyzed.

## 5. Conclusions

In this study, CDW was used for the evaluation of risk factors of POD. The occurrence of POD after hip fracture surgery was 18.7%. There was no relationship between perioperative anemia and red blood cell transfusion and the occurrence of POD in hip fracture surgery. However, postoperative red blood cell transfusion was associated with the development of POD. Therefore, red blood cell transfusion in relation to complications (like increased inflammation and oxidative stress) contributes to the occurrence of POD. Big data analytics using CDW is a good option to identify the risk factors of POD and to prevent POD in hip fracture surgery.

## Figures and Tables

**Figure 1 jcm-12-04696-f001:**
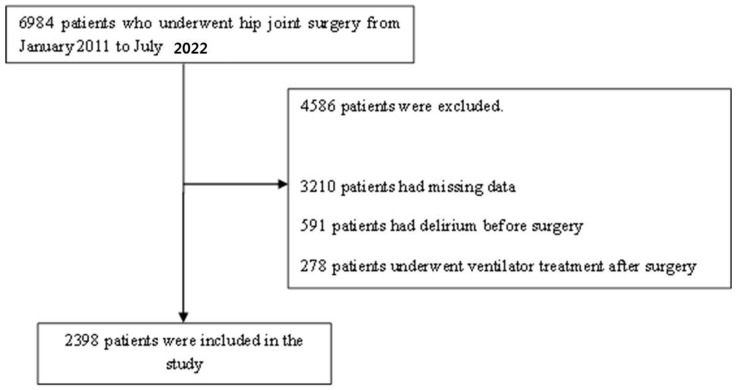
Flow chart.

**Table 1 jcm-12-04696-t001:** Preoperative, intraoperative, and postoperative factors of patients.

	No Delirium (*n* = 1950)	Delirium (*n* = 448)	*p*-Value
Preoperative factors			
Age, year (median [IQR], maximum)	76 ([65, 82], 99)	82 ([76, 87], 103)	<0.001
Male, *n* (%)	651 (33.4)	131 (29.2)	0.092
Obesity (Bodymassindex > 29.9), *n* (%)	60 (3.1)	4 (0.9)	0.01
ASA physical status >2, *n* (%)	953 (48.9)	350 (78.1)	<0.001
Emergency surgery (*n*, %)	449 (23.0)	84 (18.8)	0.05
Hypertension, *n* (%)	1067 (54.7)	290 (64.7)	<0.001
Diabetes, *n* (%)	507 (26.0)	133 (29.7)	0.112
Stroke, *n* (%)	270 (13.8)	80 (17.9)	0.03
Cancer, *n* (%)	196 (10.1)	55 (12.3)	0.165
Parkinson disease, *n* (%)	49 (2.5)	23 (5.1)	0.003
Dementia, *n* (%)	113 (5.8)	78 (17.4)	<0.001
Depression, *n* (%)	70 (3.6)	19 (4.2)	0.511
Kidney disease, *n* (%)	135 (6.9)	33 (7.4)	0.74
Liver disease, *n* (%)	89 (4.6)	17 (3.8)	0.475
Insomnia, *n* (%)	128 (6.6)	40 (8.9)	0.077
Alcohol, *n* (%)	355 (18.2)	48 (10.7)	<0.001
Smoking, *n* (%)	249 (12.8)	44 (9.8)	0.086
Calcium channel blocker, *n* (%)	757 (38.8)	199 (44.4)	0.029
Diuretics, *n* (%)	212 (10.9)	45 (10.0)	0.61
Beta blocker, *n* (%)	171 (8.8)	29 (6.5)	0.113
Angiotensin converting enzyme inhibitors, *n* (%)	15 (0.8)	5 (1.1)	0.467
Angiotensin receptor blockers, *n* (%)	131 (6.7)	32 (7.1)	0.747
Miscellaneous drug, *n* (%)	201 (10.3)	25 (5.6)	0.002
Anti-depressant, *n* (%)	58 (3.0)	8 (1.8)	0.166
Hypnotics, *n* (%)	780 (40.0)	146 (32.6)	0.004
Anti-psychotics, *n* (%)	277 (14.2)	65 (14.5)	0.868
Opioid, *n* (%)	1921 (98.5)	442 (98.7)	0.814
Non-steroid alanti-inflammatory drug, *n* (%)	549 (28.2)	93 (20.8)	0.001
Analgesic except non-steroid alanti-inflammatory drug, *n* (%)	1884 (96.6)	439 (98.0)	0.131
Muscle relaxant, *n* (%)	794 (40.7)	122 (27.2)	<0.001
Steroid, *n* (%)	188 (9.6)	24 (5.4)	0.004
Anti-platelet, *n* (%)	69 (3.5)	19 (4.2)	0.476
Anti-coagulant, *n* (%)	1326 (68.0)	358 (79.9)	<0.001
Anti-hyperlipidemic, *n* (%)	45 (2.3)	20 (4.5)	0.011
Anti-Parkinson, *n* (%)	6 (0.3)	5 (1.1)	0.022
Antihistamine, *n* (%)	826 (42.4)	267 (59.6)	<0.001
Genitourinary drug, *n* (%)	2 (0.1)	2 (0.4)	0.108
H2 blocker, *n* (%)	474 (24.3)	90 (20.1)	0.058
Preoperative hemoglobin (9.9–8/7.9–6/<6, g/dL), *n* (%)	248(12.7)/10 (0.5)/0 (0)	71(15.8)/3 (0.7)/0 (0)	0.192
Differences from normal levels of aspartate aminotransferase before surgery, units/L (median [IQR], maximum)	0 ([0, 0], 282)	0 ([0, 0], 422)	0.895
Differences from normal levels of alanine aminotransferase before surgery, units/L (median [IQR], maximum)	0 ([0, 0], 303)	0 ([0, 0], 654)	0.274
Preoperative hypoNa^+^/hyperNa^+^, *n* (%)	224 (11.5)/7 (0.4)	72 (16.1)/3 (0.7)	0.018
Preoperative hypoK^+^/hyperK^+^, *n* (%)	195 (10.0)/4 (0.2)	68 (15.2)/1 (0.2)	0.007
Differences from normal levels of uric acid before surgery, mg/dL (median [IQR], maximum)	0 ([0, 0], 6.9)	0 ([0, 0], 5.5)	0.015
Differences from normal levels of blood urea nitrogen before surgery, mg/dL (median [IQR], maximum)	0 ([0, 0], 54.8)	0 ([0, 0.1], 47.9)	0.036
Differences from normal levels of creatinine before surgery creatinine, mg/dL (median [IQR], maximum)	0 ([0, 0], 9.0)	0 ([0, 0], 5.1)	0.086
Preoperative hypoalbuminemia/hyperalbuminemia, *n* (%)	948 (48.6)/1 (0.1)	266 (59.4)/0 (0)	<0.001
Intraoperative factors			
Generalanesthesia, *n* (%)	1343 (68.9)	343 (76.6)	0.001
Surgical type (ORIF/Hemi-R/THA), *n* (%)	966(49.5)/759 (38.9)/225 (11.5)	217(48.4)/210 (46.9)/21 (4.7)	<0.001
Operation time, hour (median [IQR], maximum)	1.2 ([1, 1.7], 8.4)	1.2 ([0.9, 1.5], 4)	0.007
Intraoperative packed red blood cell, pack (median [IQR], maximum)	0 ([0, 1], 10)	1 ([0, 1], 6)	0.003
Intraoperative fresh frozen plasma, pack (median [IQR], maximum)	0 ([0, 0], 6)	0 ([0, 0], 3)	0.043
Intraoperative platelet concentration, pack (median [IQR], maximum)	0 ([0, 0], 10)	0 ([0, 0], 10)	0.357
Opioid use, *n* (%)	967 (49.6)	268 (59.8)	<0.001
Midazolam, mg (median [IQR], maximum)	0 ([0, 0], 15)	0 ([0, 0], 5)	<0.001
Oliguria (<0.5 mL/kg/hour), *n* (%)	577 (29.6)	120 (26.8)	0.239
Administered fluid, L (median [IQR], maximum)	1 ([0.8, 1.4], 6.3)	1 ([0.8, 1.3], 3.7)	0.301
Estimated blood loss, L (median [IQR], maximum)	0.5 ([0.2, 0.5], 10)	0.5 ([0.2, 0.5], 2.5)	0.269
Postoperative factors			
Postoperative packed red blood cell, pack/day (median [IQR], maximum)	0 ([0, 0.3], 1.6)	0 ([0, 0.3], 6)	0.901
Postoperative fresh frozen plasma, pack/day (median [IQR], maximum)	0 ([0, 0], 0.7)	0 ([0, 0], 4)	0.042
Postoperative platelet concentration, pack/day (median [IQR], maximum)	0 ([0, 0], 4.3)	0 ([0, 0], 5)	0.04
Postoperative hemoglobin (9.9–8/7.9–6/<6, g/dL), *n* (%), g/dL (median [IQR], maximum)	379 (19.4)/24 (1.2)/2 (0.1)	94 (21.0)/9 (2.0)/1 (0.2)	0.433
Differences from normal levels of postoperative aspartate aminotransferase, units/L (median [IQR], maximum)	0 ([0, 0], 2475)	0 ([0, 0], 178)	0.29
Differences from normal levels of postoperative alanine aminotransferase, units/L (median [IQR], maximum)	0 ([0, 0], 1401)	0 ([0, 0], 71)	0.289
Postoperative hypoNa^+^/hyperNa^+^, mEq/L (median [IQR], maximum)	248 (12.7)/22 (1.1)	84 (18.8)/6 (1.3)	0.003
Postoperative hypoK^+^/hyperK^+^, mmol/L (median [IQR], maximum)	234 (12.0)/12 (0.6)	80 (17.9)/4 (0.9)	0.003
Differences from normal levels of postoperative uric acid, mg/dL (median [IQR], maximum)	0 ([0, 0], 3.9)	0 ([0, 0], 3.8)	0.451
Differences from normal levels of postoperative blood urea nitrogen, mg/dL (median [IQR], maximum)	0 ([0, 0], 48.1)	0 ([0, 0], 58.3)	0.014
Differences from normal levels of postoperative creatinine, mg/dL (median [IQR], maximum)	0.7 (0.6, 0.9)	0.7 (0.6, 0.9)	0.212
Postoperative albumin, g/dL (median [IQR], maximum)	1866 (95.7)/0 (0)	442 (98.7)/0 (0)	0.003
Differences from normal levels in postoperative maximum body temperature, °C (median [IQR], maximum)	0.7 ([0.3, 1.1], 3)	0.7 ([0.3, 1.1], 2.5)	0.455
Differences from normal levels of postoperative erythrocyte sedimentation rate, mm/hr (median [IQR], maximum)	30 ([15, 48], 104)	15 ([2, 32], 104)	<0.001
Differences from normal levels of postoperative C-reactive protein, mg/L (median [IQR], maximum)	0 ([0, 0], 8.2)	0 ([0, 0], 8.7)	0.244
Intensive care unit, *n* (%)	255 (13.1)	114 (25.4)	<0.001
Patient-controlled analgesia, *n* (%)	1906 (97.7)	442 (98.7)	0.221

IQR, interquartile range; ASA, American Society of Anesthesiologists; hypoNa+, hyponatremia; hyperNa+, hypernatremia; hypoK+, hypokalemia; hyperK+, hyperkalemia; ORIF, open reduction and internal fixation; Hemi-R, hemiarthroplasty; THA, total hip arthroplasty.

**Table 2 jcm-12-04696-t002:** Intraoperative and postoperative red blood cell transfusion distributions.

Intraoperative RBC Transfusion, Pack			Postoperative RBC Transfusion, Pack		
Pack	No Delirium, *n* (%)	Delirium, *n* (%)	Percentile	No Delirium, Pack/Day	Delirium, Pack/Day
0	1140(58.5)	222 (49.6)	10	0	0
1	375 (19.2)	116 (25.9)	20	0	0
2	337 (17.3)	80 (17.9)	30	0	0
3	66 (3.4)	24 (5.4)	40	0	0
4	19 (1.0)	4 (0.9)	50	0	0
5	6 (0.3)	1 (0.2)	60	0.1	0
6	2 (0.1)	1 (0.2)	70	0.3	0.3
7	1 (0.1)	0 (0.0)	80	0.3	0.5
8	3 (0.2)	0 (0.0)	90	0.3	0.8
9	0 (0.0)	0 (0.0)			
10	1 (0.1)	0 (0.0)			

RBC, red blood cell.

**Table 3 jcm-12-04696-t003:** The odds ratio of intraoperative and postoperative red blood cell transfusion (pack) for developing postoperative delirium.

		Odds Ratio [95% Confidence Interval]	*p*-Value
Intraoperative packed red blood cell, pack	Unadjusted	0.94 [0.86–1.02]	0.143
	Adjusted	1.02 [0.9–1.17]	0.751
Postoperative packed red blood cell, pack/day	Unadjusted	1.28 [1.04–1.57]	0.022
	Adjusted	2.53 [1.88–3.41]	<0.001

**Table 4 jcm-12-04696-t004:** The odds ratios of preoperative, intraoperative, and postoperative risk factors for developing postoperative delirium.

	Hazard Ratio (95% Confidence Interval)	*p*-Value
Preoperative Factors		
Age, year	1.05 (1.04 to 1.07)	<0.001
Male	1.16 (0.9 to 1.5)	0.26
Obesity (Body mass index > 29.9)	0.37 (0.14 to 1.01)	0.05
American Society of Anesthesiologists physical status >2	1.84 (1.4 to 2.42)	<0.001
Emergency surgery	1.02 (0.79 to 1.32)	0.86
Hypertension	0.98 (0.79 to 1.21)	0.86
Diabetes	1.34 (1.08 to 1.67)	0.01
Stroke	1.13 (0.87 to 1.48)	0.36
Cancer	1.15 (0.85 to 1.56)	0.37
Parkinson disease	1.5 (0.94 to 2.39)	0.09
Dementia	1.99 (1.53 to 2.6)	<0.001
Depression	1.05 (0.64 to 1.71)	0.85
Kidney disease	0.89 (0.57 to 1.39)	0.62
Liver disease	0.68 (0.4 to 1.16)	0.16
Insomnia	1.13 (0.79 to 1.61)	0.5
Alcohol	1.21 (0.84 to 1.76)	0.31
Smoking	1.43 (0.98 to 2.1)	0.06
Calcium channel blocker	1.05 (0.85 to 1.29)	0.68
Diuretics	0.9 (0.63 to 1.28)	0.55
Beta blocker	0.71 (0.47 to 1.07)	0.1
Angiotensin converting enzyme inhibitors	0.68 (0.25 to 1.82)	0.44
Angiotensin receptor blockers	1.25 (0.85 to 1.85)	0.26
Miscellaneous drug	0.65 (0.41 to 1.03)	0.07
Anti-depressant	0.84 (0.4 to 1.77)	0.65
Hypnotics	0.71 (0.55 to 0.91)	0.01
Anti-psychotics	1.29 (0.95 to 1.76)	0.11
Opioid	0.58 (0.22 to 1.5)	0.26
Non-steroid alanti-inflammatorydrug	0.74 (0.58 to 0.94)	0.01
Analgesic except non-steroidalanti-inflammatorydrug	1.63 (0.8 to 3.29)	0.18
Muscle relaxant	0.75 (0.59 to 0.95)	0.02
Steroid	0.53 (0.34 to 0.82)	<0.001
Anti-platelet	0.93 (0.55 to 1.58)	0.8
Anti-coagulant	1.07 (0.81 to 1.43)	0.62
Anti-hyperlipidemic	1.34 (0.81 to 2.22)	0.26
Anti-Parkinson	1.22 (0.43 to 3.44)	0.71
Antihistamine	1.53 (1.19 to 1.96)	<0.001
Genitourinary drug	1.18 (0.26 to 5.4)	0.83
H2 blocker	0.9 (0.69 to 1.18)	0.45
Preoperative hemoglobin		
9.9–8 g/dL	0.86 (0.64 to 1.16)	0.32
7.9–6 g/dL	0.8 (0.23 to 2.84)	0.73
Differences from normal levels of aspartate aminotransferase before surgery, units/L	1.0 (0.99 to 1.02)	0.92
Differences from normal levels of alanine aminotransferase before surgery, units/L	1.01 (1.0 to 1.02)	0.19
Preoperative Sodium		
Hyponatremia	0.95 (0.71 to 1.27)	0.73
Hypernatremia	1.14 (0.3 to 4.43)	0.85
Preoperative Potassium		
Hypokalemia	1.3 (0.96 to 1.76)	0.09
Hyperkalemia	1.07 (0.14 to 7.9)	0.95
Differences from normal levels of uric acid before surgery, mg/dL	1.17 (0.92 to 1.5)	0.2
Differences from normal levels of blood urea nitrogen before surgery, mg/dL	1.0 (0.97 to 1.03)	0.83
Differences from normal levels of creatinine before surgery creatinine, mg/dL	0.75 (0.45 to 1.26)	0.28
Preoperative Hypoalbuminemia	0.94 (0.76 to 1.17)	0.59
Intraoperativefactors		
Generalanesthesia	0.92 (0.69 to 1.22)	0.55
Surgical type (reference: ORIF)		
Hemiarthroplasty	0.97 (0.78 to 1.22)	0.81
Totalarthroplasty	0.97 (0.59 to 1.59)	0.9
Operation time, hour	1.01 (0.81 to 1.26)	0.93
Intraoperative fresh frozen plasma, pack	0.87 (0.51 to 1.48)	0.6
Intraoperative platelet concentration, pack	1.11 (0.93 to 1.31)	0.24
Opioid use	1.04 (0.82 to 1.32)	0.75
Midazolam, mg	1.01 (0.85 to 1.2)	0.9
Oliguria (<0.5 mL/kg/hour)	1.03 (0.81 to 1.31)	0.8
Administered fluid, L	0.91 (0.68 to 1.22)	0.54
Estimated blood loss, L	0.96 (0.62 to 1.5)	0.87
Postoperative factors		
Postoperative fresh frozen plasma, pack/day	0.39 (0.2 to 0.77)	0.01
Postoperative platelet concentration, pack/day	0.77 (0.51 to 1.18)	0.23
Postoperative hemoglobin		
9.9–8 g/dL	0.83 (0.64 to 1.07)	0.15
7.9–6 g/dL	0.93 (0.45 to 1.93)	0.86
<6 g/dL	1.48 (0.18 to 12.17)	0.71
Differences from normal levels of postoperative aspartate aminotransferase, units/L	1.01 (1.0 to 1.02)	0.32
Differences from normal levels of postoperative alanine aminotransferase, units/L	0.96 (0.91 to 1.02)	0.21
Postoperative sodium		
Hyponatremia	1.08 (0.81 to 1.43)	0.61
Hypernatremia	1.29 (0.56 to 2.99)	0.55
Postoperative potassium		
Hyponatremia	1.13 (0.85 to 1.5)	0.41
Hypernatremia	1.48 (0.48 to 4.51)	0.5
Differences from normal levels of postoperative uric acid, mg/dL	0.96 (0.56 to 1.63)	0.87
Differences from normal levels of postoperative blood urea nitrogen, mg/dL	1.0 (0.97 to 1.03)	0.96
Differences from normal levels of postoperative creatinine, mg/dL	1.27 (0.85 to 1.9)	0.24
Postoperativehypoalbuminemia	0.86 (0.37 to 2.02)	0.74
Differences from normal level in postoperative maximum body temperature, °C	0.92 (0.75 to 1.14)	0.46
Differences from normal levels of postoperative erythrocyte sedimentation rate, mm/h	0.97 (0.97 to 0.98)	<0.001
Differences from normal levels of postoperative C-reactive protein, mg/L	1.0 (1.0 to 1.0)	0.25
Intensive care unit	1.31 (1.02 to 1.68)	0.03
Patient-controlled analgesia	0.95 (0.4 to 2.25)	0.91

## Data Availability

All data were obtained from the clinical data warehouse (CDW) of the five hospitals of Hallym University Medical Center.

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
