# Peer review of "The Impact of Blood Transfusion in Developing Postoperative Delirium in Patients with Hip Fracture Surgery"

_jcm, 2023, doi:10.3390/jcm12144696_

Round 1

Reviewer 1 Report

Comments on Manuscript: jcm-2446345

Title: The impact of blood transfusion in developing postoperative delirium in patients with hip fracture surgery. 

The authors present their study on the identification of risk factors of post-op delirium (POD) among patients undergoing hip fracture surgery, specifically evaluating the impact of blood transfusions.

The paper is adequately written and organized and the topic is interesting.

I think that some adjustments should be performed.

Comments:

- Dealing with POD in elderly patients, it is rather useless to consider as inclusion criteria “patients of 18 years or older”: conventionally, patients over 65 years of age are considered included in such these studies. So, also the flow chart should not start from a >4000 cases included and then be removed if, according to the results, only subjects >65 years of age were considered.

- Table 1 is frankly unreadable and hard to comprehend: please try to improve such crucial data.

no specific comments

Reviewer 2 Report

I read with great interest the manuscript by Lee et al. On The impact of blood transfusion in developing postoperative delirium in patients with hip fracture surgery. The paper is original and well written. However, I have some issues to be addressed:

- Line 18. Please replace “surgeriesical” with “surgical”.

- Line 37. Authors should add that several strategies have been studied to prevent and treat POD, with mixed results (doi: 10.2147/CIA.S201323 - doi: 10.3390/jcm12020435 - doi: 10.1016/j.ccc.2017.03.013). Please briefly discuss and add these 3 references.

- Line 41. Please replace “caused by intraoperative blood loss” with “caused by hypotensive events related to intraoperative blood loss”. 

- Please delete the sentence “You can search for patients by prescription, examination, diagnosis, and more.”

- Line 71. Please delete “were eligible for analysis”

- Line 72. Please replace “are” with “were”.

- 168-171. This sentence is quite redundant. I would suggest to delete it.

- Line 201-202. Please provide a reference for this sentence.

- Line 201-202. Authors should add that in recent years, new intraoperative monitoring have proven to prevent hypotensive events using artificial intelligence (doi: 10.1007/s10877-019-00433-6 - doi: 10.3390/jcm11020392), possibly leading to an incidence reduction in the future. Please briefly discuss and add these 2 references.

Author Response

Reviewer 2.

Comments and Suggestions for Authors

I read with great interest the manuscript by Lee et al. On The impact of blood transfusion in developing postoperative delirium in patients with hip fracture surgery. The paper is original and well written. However, I have some issues to be addressed:

- Line 18. Please replace “surgeriesical” with “surgical”.

Answer: Thank for your very kindly advice. We sincerely revised our study like below.

Medical data of patients who underwent ~

- Line 37. Authors should add that several strategies have been studied to prevent and treat POD, with mixed results (doi: 10.2147/CIA.S201323 - doi: 10.3390/jcm12020435 - doi: 10.1016/j.ccc.2017.03.013). Please briefly discuss and add these 3 references.

Answer: Thank for your very kindly advice. We sincerely revised our study like below, add these 3 references.

In previous studies, several strategies (including the identification of the risk factors for POD) have been studied to identify the pathogenesis of POD and prevent POD in hip fracture surgery.

- Line 41. Please replace “caused by intraoperative blood loss” with “caused by hypotensive events related to intraoperative blood loss”.

Answer: Thank for your very kindly advice. We sincerely revised our study like below.

caused by hypotensive events related to intraoperative blood loss

- Please delete the sentence “You can search for patients by prescription, examination, diagnosis, and more.”

Answer: Thank for your very kindly advice. We delete the sentence “You can search for patients by prescription, examination, diagnosis, and more.”

- Line 71. Please delete “were eligible for analysis”

Answer: Thank for your very kindly advice. We delete the sentence.

- Line 72. Please replace “are” with “were”.

Answer: Thank for your very kindly advice. We sincerely revised our study.

- 168-171. This sentence is quite redundant. I would suggest to delete it.

Answer: Thank for your very kindly advice. We sincerely revised our study.

- Line 201-202. Please provide a reference for this sentence.

- Line 201-202. Authors should add that in recent years, new intraoperative monitoring have proven to prevent hypotensive events using artificial intelligence (doi: 10.1007/s10877-019-00433-6 - doi: 10.3390/jcm11020392), possibly leading to an incidence reduction in the future. Please briefly discuss and add these 2 references.

Answer: Thank for your very kindly advice. We sincerely revised our study like below.

Prolonged hypotension leads to low cerebral perfusion, cerebral ischemia, hypoxia, impairment of brain function [31,32]. In recent years, new monitoring-tool using artificial intelligence has been developed to predict episodes of intraoperative hypotension and prevent prolonged hypotension [33,34].   
